# *Aeromonas sobria* Induced Sepsis Complicated with Necrotizing Fasciitis in a Child with Acute Lymphoblastic Leukemia

Manuela Colosimo [1], Maria Concetta Galati [2], Umberto Riccelli [3], Simona Paola Tiburzi [4], Eulalia Galea [2], Teresa Alcaro [1], Orazio Stefano Giovanni Filippelli [4], Francesco Abbonante [3], Pasquale Minchella [1] and Luca Gallelli [5,6,*]

1 Operative Unit of Microbiology and Virology, "Pugliese Ciaccio" Hospital, 88100 Catanzaro, Italy; manuelacolosimo@hotmail.it (M.C.); teresaalcaro86@gmail.com (T.A.); minchellap@tin.it (P.M.)
2 Department of Oncology and Hematology, "Pugliese Ciaccio" Hospital, 88100 Catanzaro, Italy; mcgalati@aocz.it (M.C.G.); eulgal@yahoo.it (E.G.)
3 Operative Unit of Plastic Surgery, Department of Surgery, "Pugliese Ciaccio" Hospital, 88100 Catanzaro, Italy; umbertoriccelli@hotmail.it (U.R.); franco.abbonante@gmail.com (F.A.)
4 Department of Critical Care and Intensive Unit, "Pugliese Ciaccio" Hospital, 88100 Catanzaro, Italy; tiburzi1@virgilio.it (S.P.T.); oraziofilippelli@alice.it (O.S.G.F.)
5 Clinical Pharmacology Unit "Mater Domini" Hospital, Department of Health Science, University Magna Graecia of Catanzaro, 88100 Catanzaro, Italy
6 Research Center FAS@UMG, Department of Health Science, University Magna Graecia of Catanzaro, 88100 Catanzaro, Italy
* Correspondence: gallelli@unicz.it; Tel.: +39-096-171-2322

**Abstract:** *Aeromonas* species are gram negative and able to induce systemic diseases (i.e., gastrointestinal, respiratory, and cardiovascular, diseases, in addition to infection of brain and soft tissues). In this study, we describe the development of necrotizing fasciitis in a young immunocompromised girl, with a low response to drug treatment and who died after some months.

**Keywords:** *Aeromonas* infection; young child; immunocompromised girl

## 1. Introduction

*Aeromonas* species are gram negative, non-sporulating facultative anaerobic bacilli that produce β-lactamases [1], able to colonize both humans and animals [2]. *Aeromonas* is ubiquitous in water, including chlorinated drinking water, where flagella facilitate biofilm formation [3]. The virulence of *Aeromonas* is multifactorial, due to cytotoxic, cytolytic, hemolytic, and enterotoxin proteins [4,5]. In human infections, the most common clinical isolates are *Aeromonas hydrophila*, *Aeromonas caviae*, and *Aeromonas veronii biovar sobria* [6], that induce systemic diseases (i.e., gastrointestinal, respiratory, and cardiovascular diseases, in addition to infection of brain and soft tissues) [7–9]. *Aeromonas* infections can develop in healthy and trauma patients, but immunocompromised hosts with hematological malignancies or solid tumors are considered to be at greatest risk [10].

In this study, we describe the development of necrotizing fasciitis in a young immunocompromised girl.

## 2. Case Presentation Section

We report a 6-year-old Caucasian girl with acute lymphoid leukemia that developed a necrotizing fasciitis related to *Aeromonas veronii biovar sobria* infection.

On 28 July, 2019 the patient was admitted to the Operative Unit of pediatric haemato-oncology with a diagnosis of acute lymphoid leukemia (EGIL B-II subtype) to start the standard chemotherapy protocol (Table 1).

**Table 1.** Chemotherapy protocol used for acute lymphoid leukemia (EGIL B-II subtype). AEIOP ALL (Associazione Italiana di Ematologia e Oncologia Pediatrica—acute lymphoid leukemia) 2017.

| Date (2019) | Phase | Cycle | Drugs |
| --- | --- | --- | --- |
| 16 January | Induction | | edoxaban plus 6-mercaptopurine |
| 18 February | | First | cytosine arabinoside |
| 1 April | | Last | |
| 18 April | Consolidation | | 6-mercaptopurine |
| 25 April | | First | Methotrexate |
| 11 June | | Last | |
| 8 July | Induction | | Dexamethasone |
| 15 July | | First | Vincristine + Adriamycin + pegaspargase |
| 22 July | | Last | |

The patient was hospitalized for clinical evaluation and laboratory findings revealed a severe decrease in blood cells (red cells $3.37 \times 10^3$/mcl, normal range 4.2–6.1; white cells $1.17 \times 10^3$/mcl, normal range 4.6–10.20; hemoglobin 10.5 gr/dL, 12–18; platelets $119 \times 10^3$/mcl, normal range 130–400; prothrombin time 68%, normal range 80–120) and an increase in liver enzymes (GPT ALT 97 IU/L, normal range 0–41; azotemia 54 g/dL, normal range 10–50) (see Table 2). Therefore, on 29 July, dexamethasone (2 mg/day orally) was added. On 30 July, the patients developed fever (39 °C) with bilateral ankle and knee fleeting erythema that disappeared within 1 h. Biochemical analysis revealed an increase in liver enzymes (GPT 139 IU/L; GOT 56 IU/L normal range 0–38), and a severe decrease in blood cell counts (red cells $2.1 \times 10^3$/mcl; white cells $0.6 \times 10^3$/mcl; platelets $50.2 \times 10^3$/mcl).

Therefore, platelets were transfused, acetaminophen (10 mL/day os) was administered, and a blood sample was taken for microbiological evaluation.

On 31 July, the patient referred to severe pain in the right upper limb (visual analogue scale, VAS, 8) with an increase in body temperature (37.7 °C), that worsened 1 day later (01 August) (fever 39 °C; VAS 10). Therefore, treatment with acetaminophen (10 mL/day os) and tramadol (8 gtt) was started, without clinical benefit. Clinical parameters were in the normal range (blood pressure, BP, 90/50 mmHg; SPO2 97%; heart frequency 160 beats/min); echo-color doppler and chest X-rays excluded phlebitis, deep venous thrombosis, or pulmonary diseases. Laboratory findings confirmed the decrease in blood cell counts (red cells $3.37 \times 10^3$/mcl; white cells $0.47 \times 10^3$/mcl; platelets $23.9 \times 10^3$/mcl).

On 02 August, a new clinical evaluation revealed the presence of fever (38.6 °C) and severe pain (VAS 10), edema, and skin rash of the right upper limb. Laboratory testing revealed a decrease in liver function and plasma proteins, and an increase in procalcitonin levels (Table 2). Microbiology diagnosed a sepsis caused by *Aeromonas veronii* biovar resistant to ampicillin (minimal inhibitory concentration (MIC) > 32), but sensitive to ciprofloxacin (MIC < 0.06), ceftriaxone (MIC < 0.25), cefuroxime (MIC < 1), trimethoprim/sulphametoxazole (MIC < 20), and ceftazidime (MIC < 0.12). An antimicrobial, antiviral, and antimycotic empirical treatment plus acetaminophen (10 mL os) and ketorolac (10 mg i.m.) was started without clinical benefit.

**Table 2.** Time course of biochemical parameters. GOT: glutamic oxaloacetic transaminase; gamma-GT: gamma-glutamyl transpeptidase; INR: international normalized ratio; LDH: lactate dehydrogenase.

| | July | August | | | | | | | | | | Normal Range | Unit |
|---|---|---|---|---|---|---|---|---|---|---|---|---|---|
| Parameters | 28 | 2 | 3 | 8 | 11 | 13 | 16 | 20 | 25 | 27 | 28 | | |
| Red blood cells | 3.37 | 2.73 | 2.83 | 3.44 | 3.63 | 3.37 | 2.81 | 2.35 | 2.9 | 2.8 | 5.4 | 4.6–10.20 | $10^3/\mu$L |
| White blood cells | 1.17 | 0.04 | 0.07 | 1.51 | 7.61 | 6.53 | 6.19 | 3.65 | 4.75 | 5.4 | 2.79 | 4.2–6.1 | $10^6/\mu$L |
| Platelets | 119 | 8.67 | 20 | 62 | 38 | 39 | 25 | 12 | 11 | 43 | 43 | 130–400 | $10^3/\mu$L |
| Hemoglobin | 10.5 | 8.2 | 8.5 | 9 | 10.9 | 10.3 | 8.8 | 7.1 | 8.7 | 8.2 | 8.2 | 012–018 | gr/dL |
| Creatinine | 0.4 | 0.2 | 0.4 | 1.3 | 0.7 | 1.1 | 0.8 | 0.7 | 0.5 | 0.7 | 0.7 | 0.5–1.2 | mg/dL |
| GOT | 36 | 111 | 166 | 9 | 15 | 14 | 22 | 17 | 16 | 16 | 24 | 0–38 | IU/L |
| Gamma-GT | 30 | 58 | 97 | 110 | 126 | 120 | 90 | 85 | 70 | 50 | 50 | 005–36 | IU/L |
| LDH | 220 | 267 | 327 | 157 | 220 | 389 | 463 | 495 | 442 | 640 | 726 | 135–250 | IU/L |
| Azotemia | 54 | 20 | 30 | 35 | 40 | 83 | 105 | 122 | 171 | 170 | 166 | 010–50 | gr/dL |
| Total Bilirubin | 0.32 | 0.7 | 3.01 | 1.64 | 1.2 | 1.08 | 0.88 | 0.75 | 0.8 | 0.9 | 0.8 | 0–1.2 | mg/dL |
| Albumin | 3.5 | 2.2 | 2.6 | 3.1 | 3.2 | 3.5 | 4.1 | 3.5 | 4.3 | 4.3 | 4.3 | 3.5–5.5 | gr/dL |
| Total protein | 4.1 | 4.4 | 4.6 | 5 | 5 | 5 | 5.2 | 5.6 | 6.9 | 7 | 7 | 6–8.7 | gr/dL |
| INR | 1.12 | 1.23 | 1.24 | 1.46 | 1.72 | 1.29 | 1.25 | 1.46 | 1.24 | 1.3 | 1.32 | 0.80–1.20 | |
| Prothrombine Time | 68 | 54.3 | 58.6 | 42.9 | 37.2 | 50.7 | 52.9 | 42.8 | 42.5 | 49 | 49 | 80–120 | % |
| Firinogen | 72 | 403 | 560 | 266 | 338 | 317 | 293 | 347 | 226 | 198 | 249 | 200–450 | mg/dL |
| Procalcitonin | | 46.2 | 44.35 | 24.7 | 21.18 | 8.7 | 7.79 | 7.5 | 7.3 | 7.2 | 8 | <0.05 | ng/mL |

On 03 August, a clinical impairment was observed, and a swab of the skin rash was sent for a new microbiology evaluation. Abdominal ultrasound was normal, whereas the echo-color Doppler of the right upper limb showed a diffuse edema of skin soft tissues without signs of thrombosis with normal function of the venous tract. Blood chemical testing confirmed the low levels of red cells with a decrease in liver function (Table 2). Blood gas analysis documented a severe respiratory failure (PO2 33 mmHg; PCO2 32 mmHg) therefore the patient was transferred to the intensive care unit (ICU).

In the ICU, the right upper limb appeared marbled with edema and severe pain (VAS 10). Dexamethasone was discontinued and a treatment with furosemide, piperacillin + tazobactam, acyclovir, clindamycin, and oxygen was started.

On 04 August, clinical evaluation documented that the patient was in comatose status and an infection sustained by *Klebsiella* species was detected (Table 3). On 06 August, the fasciotomy of the right limb revealed the presence of a black fluid where microbiology testing detected the presence of *Aeromonas veronii biovar sobria* (resistant to ampicillin, MIC > 32, but sensitive to ceftazidime MIC < 0.12, cefuroxime MIC < 1, ceftriaxone MIC < 0.25, and ciprofloxacin MIC < 0.06). On physical examination, her lower limb was warm in the presence of erythema. After 12 h, the swelling of her leg and the progression of skin lesions (circumferential erythema developed in bullae formations) increased until the development of a compartment syndrome, which rapidly required a surgical treatment with fasciotomy and negative pressure wound therapy (Figure 1).

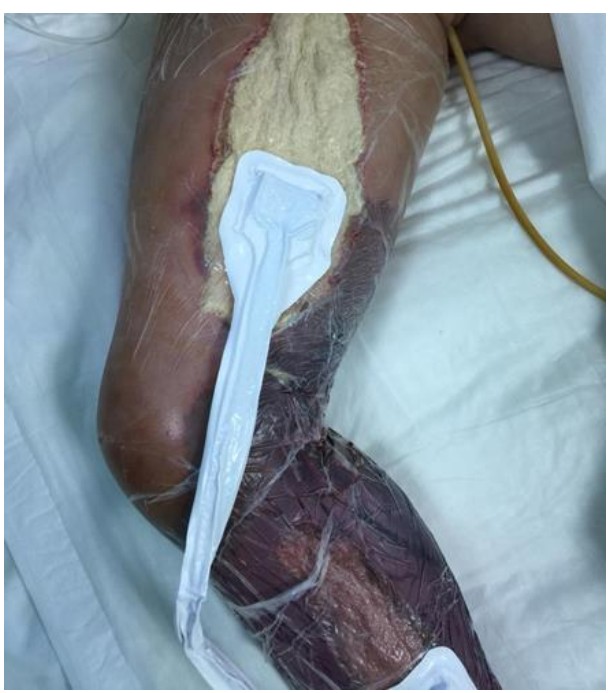

**Figure 1.** Surgical treatment with fasciotomy in young child.

**Table 3.** *Klebsiella* species isolated from swab surgical wound of the right lower limb on 04 August 2019. S: sensitive; R: resistant.

| Drugs | MIC (µg/mL) | Effect |
|---|---|---|
| Amikacin | <1 | S |
| Ampicillin | >32 | R |
| Amoxicillin/Clavulanate | >32 | R |
| Cefepime | <0.12 | S |
| Cefotaxime | <0.25 | S |
| Ceftazidime | <0.25 | S |
| Cefuroxime | <1 | S |
| Ciprofloxacin | <0.06 | S |
| Ertapenem | 0.5 | S |
| Fosfomycin | <0.16 | S |
| Gentamycin | <1 | S |
| Levofloxacin | <0.12 | S |
| Piperacillin/tazobactam | <4 | S |

On 19 August, microbiological evaluation of surgical amputation of the distal lower limb (Figure 2) documented a positivity for *Enterococcus faecium* multi-resistant (ampicillin MIC > 32; Amoxicillin/clavulanate MIC > 32; streptomycin MIC SYN-R; Teicoplanin MIC > 32; Vancomycin MIC > 32; Linezolid MIC 2; Tigecycline MIC < 0.12).

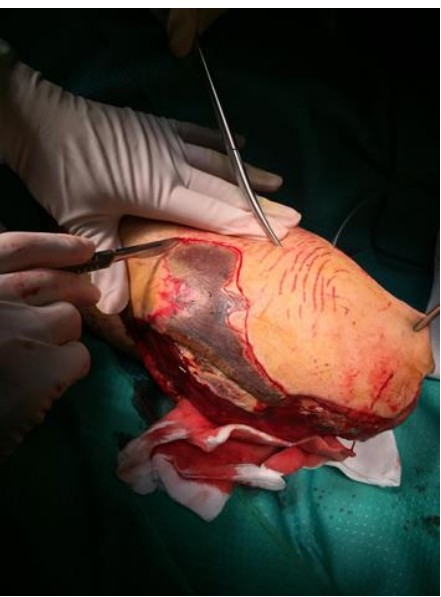

**Figure 2.** Surgical treatment with limb amputation.

On 22 August, microbiological evaluation of urine sample documented an infection sustained by the *Acinetobacter Bahmani* complex (Table 4); therefore, tigecycline (30 mg/12 h) was added to the therapy, but after 7 days, due to both the progression of leukemia and the presence of a severe antimicrobial resistance infection (Table 4), the child died.

**Table 4.** *Acinetobacter Bahmani* complex infection in urine sample on 22 August 2019. S: sensitive; R: resistant.

| Drugs | MIC (µg/mL) | Effect |
|:---:|:---:|:---:|
| Amikacin | >16 | R |
| Ampicillin | >8 | R |
| Aztreonam | >4 | R |
| Cefazoline | >16 | R |
| Cefepime | 8 | R |
| Cefixime | >1 | R |
| Cefotaxime | >32 | R |
| Cefoxitin | >16 | R |
| Ceftazidime | >32 | R |
| Cefuroxime | >8 | R |
| Ciprofloxacin | >1 | R |
| Ertapenem | >1 | R |
| Fosfomycin | 64 | R |
| Gentamycin | >4 | R |
| Imipenem | >8 | R |
| R Levofloxacin | R>1 | R |
| Mecilliname | >8 | R |
| Meropenem | >32 | R |
| Norfloxacin | >1 | R |
| Piperacillin | >16 | R |
| Ticarcillin | >16 | R |
| Tobramycin | >4 | R |
| Trimethoprim /sulphametoxazole | >4/16 | R |
| Trimethoprim | >4 | R |

## 3. Discussion

In this paper, we report an uncommon case of *Aeromonas sobria* sepsis with necrotizing fasciitis in an immunocompromised child. In a previous manuscript, Zhang et al. [11] evalu-

ated 15 patients globally (median age 49.5 years; range: 27–80 years) and documented that, in patients with acute myeloid leukemia or acute lymphoblastic leukemia and *Aeromonas sobria* infection, the most common manifestation was the lower limb infection (66.7%). In this study, the authors reported that even if an empirical antibiotic treatment is started, the mortality is high. In our case presentation, we describe the development of sepsis and lower limb infection in a young child (6 years) with acute lymphoblastic leukemia. The patient developed a first-time sepsis and then lower limb infection despite starting empirical antimicrobial treatment. This effect could be related to the co-administration of corticosteroid and chemotherapy. To date, no definitive data have been published regarding the correlation between glucocorticoid treatment and systemic infection. However, Fardet et al. [12], based on analysis of a primary care database, documented that glucocorticoid use can induce the development of respiratory infections, even if they were not able to evaluate the role of chemotherapy or biological drugs in these infections. In our study, chemotherapy probably increased the risk of septicemia by *Aeromonas sobria* with the development of lower limb infection. Radiological chest and ultrasound excluded the presence of other systemic diseases, and swabs excluded the presence of infection in other sites and, after a few days, fasciculitis induced a systemic impairment with severe respiratory failure and immune depression. Systemic antimicrobial therapy did not reduce the infection. Both fasciotomy and amputation of the lower limb improved the symptoms, but after a few days, the clinical condition worsened. Microbiology detected a multi-resistant *Aeromonas* in tissues and, despite antimicrobial treatment being started, the patient died.

## 4. Conclusions

In conclusion, this study suggests that the risk of lower limb infection in the presence of *Aeromonas* infection must be carefully evaluated, in order to quickly start an appropriate antimicrobial treatment.

**Author Contributions:** Conceptualization, M.C. and U.R.; methodology, M.C.G., T.A., E.G.; investigation, M.C., S.P.T., O.S.G.F.; resources, F.A., P.M.; data curation, M.C., E.G., S.P.T.; writing—original draft preparation, M.C.; writing—review and editing, L.G.; supervision, F.A., P.M., L.G. All authors have read and agreed to the published version of the manuscript

**Funding:** This research received no external funding.

**Institutional Review Board Statement:** The study was conducted according to the guidelines and the study was approved by the local institutional review board.

**Informed Consent Statement:** Written informed consent has been obtained from the parents of the child.

**Conflicts of Interest:** The authors declare no conflict of interest.

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
