# Peer review of "Aeromonas sobria Induced Sepsis Complicated with Necrotizing Fasciitis in a Child with Acute Lymphoblastic Leukemia"

_reports, doi:10.3390/reports4020009_

Round 1

Reviewer 1 Report

The authors reported a rare case that can be published because it is very interesting and useful for clinicians. However, major revision of the manuscript is required before publication. 

  1. Replace through the whole manuscript the informal words (ex SO, BETTER) with formal (such as – As a consequence, in this respect etc)
  2. Use clinical terms through the whole manuscript

History revealed – Medical history / PATIENT HISTORY

arrived to our observation

revealed the presence a mild increase in body temperature (37.7°C)

  1. Check the English language and style
  2. The Table 1 and 2 presented the antibiograms results. Klebsiella species isolated from swab surgical wound of the right down limb on 04 august 111 2019. S: Sensible; R: Resistant. Acinetobacter Bahmani complex infection in urine sample on 22 August 2019. S: Sensible; 113 R: Resistant. The information from the manuscript are not supported by the Table results.

Clinical evaluation excluded other systemic diseases and 50 laboratory findings revealed a severe decrease in blood cells (red cells 3.37*103/mcl; 51 normal range 4.2-6.1; white cells 1.17*103/mcl normal range 4.6-10.20; hemoglobin 10.5 52 gr/dL, 12-18; platelets 119 *103/mcl; normal range, 130-400; prothrombine time 68%; 53 nomal range 80-120) and an increase in liver enzymes (GPT ALT 97 IU/L, normal range 0- 54 41; azotemia 54 g/dL, normal range10-50) (see Table 1)

Laboratory findings confirmed the decrease in blood cells parameters (Table 1).

Laboratory test revealed a decrease in liver function and in plasma proteins and an increase in procalcitonin levels (Table 1).

Blood chemical test confirmed the low levels of red cells with a decrease in liver function (Table 1).

On 04 August the clinical conditions were impaired (Table 1), the patient was in comatose status,

Microbiological evaluation of urine sample documented an infection sustained by Acinetobacter Bahmani complex multi-resistant (Table 1)

after 10 days the children died to the progression of leukemia (table 2).

  1. LLA- Use abbreviation at first appearance. If the authors decided to use abbreviations use ALL for Acute lymphoblastic leukemia
  2. The manuscript need updated references

Author Response

Reviewer 1

The authors reported a rare case that can be published because it is very interesting and useful for clinicians. However, major revision of the manuscript is required before publication. 

Dear Reviewer,

 thank you for you comments that we take in consideration to improve our manuscript. We send you the new version and the answers to your comments

  1. Replace through the whole manuscript the informal words (ex SO, BETTER) with formal (such as – As a consequence, in this respect etc)
  2. Use clinical terms through the whole manuscript

These changes have been performed in this new version of our manuscript

  1. Check the English language and style

Manuscript has been revised for English language and style

  1. The Table 1 and 2 presented the antibiograms results. Klebsiella species isolated from swab surgical wound of the right down limb on 04 august 111 2019. S: Sensible; R: Resistant. Acinetobacter Bahmani complex infection in urine sample on 22 August 2019. S: Sensible; 113 R: Resistant. The information from the manuscript are not supported by the Table results.

Effectively there was an error in our tables. These have been revisited, table 1 and 2 have been added.

Reviewer 2 Report

Colosimo et al have described a case of a 6-year-old Caucasian girl with acute lymphoid leukemia that developed necrotizing fasciitis related to Aeromonas veronii biovar sobria infection.

The authors have described the case very well. I have the following suggestions to make the manuscript better?

It would be better to present the patient's baseline history like WBC count, Subtype of ALL, chest X-ray, and other tests which are done before starting chemotherapy.

The patient is a girl but she is referred to as he in the manuscript (line 91-98). Please check it. 

The authors did mention that parents have provided the written consent to publish their daughter's data. I am just wondering whether they provided any evidence of it. 

Thank you

Author Response

Reviewer 2

Dear Reviewer,

thank you for the comments that we take in consideration to improve our manuscript. We send you the new version and the answers to your comments

  • It would be better to present the patient's baseline history like WBC count, Subtype of ALL, chest X-ray, and other tests which are done before starting chemotherapy.

A Table regarding the baseline characteristics of the patients before the enrollment in our division and during the hospitalization have been added. Unfortunately, we have not the data requested because she was treated in a different hospital.

  • The patient is a girl but she is referred to as he in the manuscript (line 91-98). Please check it.

Thanks for this observation, manuscript has been revised for English language and style

  • The authors did mention that parents have provided the written consent to publish their daughter's data. I am just wondering whether they provided any evidence of it.

We have it and routinely, we request it for all patient (or parents) and it is enclosed in medical record. This history has not been easy for the parents.

Round 2

Reviewer 1 Report

Delete the second appearance of ''On July 28, 2019'' on the second page, continue writing that the patient was hospitalized for clinical evaluation. 

Add ''As a''  ''consequence, platelets were transfused....''

Author Response

dear Reviewer 

thank you for your comments,

 we revised the manuscript according these.

In particular

- the sentence "On July 28, 2019'' on the second page was deleted

- "As a" was added

- English form was reviewed again